published: 25 August 2021

# Cost-Effectiveness of a School-and Family-Based Childhood Obesity Prevention Programme in China: The "CHIRPY DRAGON" Cluster-Randomised Controlled Trial

Mandana Zanganeh[1], Peymane Adab[2], Bai Li[3], Miranda Pallan[2], Wei Jia Liu[4]*, Karla Hemming[2], Rong Lin[4], Wei Liu[4], James Martin[2], Kar Keung Cheng[2] and Emma Frew[2]*

[1]Warwick Medical School, University of Warwick, Coventry, United Kingdom, [2]Institute for Applied Health Research, University of Birmingham, Birmingham, United Kingdom, [3]School for Public Studies, University of Bristol, Bristol, United Kingdom, [4]Guangzhou Center for Disease Control and Prevention, Guangzhou, China

**Objectives:** Rapid socioeconomic and nutrition transitions in Chinese populations have contributed to the growth in childhood obesity. This study presents a cost-effectiveness analysis of a school- and family-based childhood obesity prevention programme in China.

**Methods:** A trial-based economic evaluation assessed cost-effectiveness at 12 months. Forty schools with 1,641 children were randomised to either receive the multi-component (diet and physical activity) intervention or to continue with usual activities. Both public sector and societal perspectives were adopted. Costs and benefits in the form of quality-adjusted life years (QALYs) were compared and uncertainty was assessed using established UK and US thresholds.

**Results:** The intervention cost was 35.53 Yuan (£7.04/US$10.01) per child from a public sector perspective and 536.95 Yuan (£106/US$151) from a societal perspective. The incremental cost-effectiveness ratio (ICER) was 272.7 Yuan (£54/US$77)/BMI z-score change. The ICER was 8,888 Yuan (£1,760/US$2,502) and 73,831 Yuan (£14,620/US$20,796) per QALY from a public sector and societal perspective, respectively and was cost-effective using UK (£20,000) and US (US$50,000) per QALY thresholds.

**Conclusion:** A multi-component school-based prevention programme is a cost-effective means of preventing childhood obesity in China.

Keywords: children, prevention, obesity, China, economic evaluation, school based intervention

**Edited by:**
Erica Di Ruggiero,
University of Toronto, Canada

**\*Correspondence:**
Wei Jia Liu
liuweijia888@126.com
Emma Frew
e.frew@bham.ac.uk

**Citation:**
Zanganeh M, Adab P, Li B, Pallan M, Liu WJ, Hemming K, Lin R, Liu W, Martin J, Cheng KK and Frew E (2021) Cost-Effectiveness of a School-and Family-Based Childhood Obesity Prevention Programme in China: The "CHIRPY DRAGON" Cluster-Randomised Controlled Trial. Int J Public Health 66:1604025. doi: 10.3389/ijph.2021.1604025

## INTRODUCTION

Childhood obesity is a global public health problem with associated health, social, and emotional consequences, as well as long term direct and indirect costs [1]. Compared to other countries, China has experienced more rapid socioeconomic and nutritional transitions in urban populations which has contributed to the rising prevalence of overweight/obesity among children [2]. The prevalence of overweight and obesity combined in school-aged children increased from around 1% (in both genders) in 1985 to 28.2% in boys and 16.4% in girls in 2015 [3]. It is therefore vital to develop (cost−) effective, culturally appropriate, obesity prevention interventions in China to curb this growing epidemic.

According to the World Health Organisation (WHO), obesity is rising in every region of the world including within low and middle income countries (LMICs) [4]. Very few obesity intervention studies have been conducted within countries with a similar economic and demographic profile to China [5]. Some trials of obesity prevention interventions implemented in a school setting in China reported effective outcomes [6–8]. However, within a recent review, none of these trials were rated high methodological quality, mainly because of selection bias and methodological limitations such as non-blinded assessments, not reporting dropouts, insufficient adjustment for confounders, and not using intention to treat analysis [6–8]. Furthermore, most studies were treatment focused and used a non-randomised design [8]. Moreover, the cost-effectiveness evidence remains unclear. Officially, China is classed as an upper-middle-income country but its per capita income remains a quarter of that of high-income countries and a large proportion of the Chinese population live below the upper-middle-income poverty line of US$5.50 a day [9]. Like many other countries, China suffers from a scarcity of public health resources and decision makers need to prioritise spending towards policies that offer the greatest value for money. Economic evaluation is a means to aid decisions about public resource allocation [10] and as obesity prevention often involves intervention or policy with costs and consequences that fall outside the health care sector, a societal perspective for evaluation is recommended to ensure these wider costs and effects are included [10].

To address the above gaps, The Chinese Primary School Children Physical Activity and Dietary Behaviour Changes Intervention (CHIRPY DRAGON) study was conducted [11]. This was a randomised controlled trial designed using the guidelines from the United Kingdom Medical Research Council (MRC) framework for complex interventions [12], to assess the clinical and cost-effectiveness of the CHIRPY DRAGON intervention. The trial randomly recruited 40 schools from urban districts of Guangzhou, China and found a mean difference in BMI z score (intervention—control) of −0.13 (−0.26 to 0.00, $p$ = 0.048) after a 12 month intervention that included both school- and family-based physical activity and dietary components [11]. This paper extends this trial analysis by reporting detailed cost-effectiveness results from both a public sector and a societal perspective to capture the costs and outcomes that fall onto the schools (public sector) as well as other sectors of the economy (community: family members). Furthermore, it measures the costs and the health and quality of life of the children and household members, and therefore captures any household spillover effects from the school-based intervention. To assess the uncertainty within the results, extensive sensitivity analysis is also reported. The methods and results are reported using consolidated health economic evaluation reporting standards (CHEERS) guidelines [13] (see **Supplementary Table SA** within the supplementary file).

## METHODS

### Trial Design

Detailed information about the trial has been reported elsewhere [11, 14]. In brief, the cluster randomised controlled trial (cRCT) evaluated the CHIRPY DRAGON obesity prevention intervention, for children aged 6–7 years at baseline, in 40 state-funded primary schools in Guangzhou, China. Schools were randomly allocated to either the usual practice ($n$ = 20) or intervention arm ($n$ = 20). In China, primary schools have an average of four (range two-eight) year-one classes per school. Each class consists of around 45 children. In the participating schools, all year-one children, along with family members, were eligible for inclusion and were offered the opportunity to take part in the intervention. One year-one class was randomly selected from each of the schools to have outcome measurements taken. However, the school-based intervention components were delivered to all classes of year one. The programme was a 12 month multi-component intervention implemented from March 2016 to March 2017. It consisted of four core components targeting diet and physical activity behaviours, inside and outside of school, through improvement of school lunch and physical-activity provisions (improving the school environment), interactive workshops targeting health knowledge and practical skills among parents, grandparents and children; and daily healthy behavioural challenges at home with individual goal setting and feedback [11]. The development of the intervention programme was guided by the MRC framework for intervention development and evaluation with application of Social Marking principles and Behaviour Change Techniques [15]. The intervention was delivered by five full-time Chinese project staff (known as CHIRPY DRAGON teachers). Each of the five CHIRPY DRAGON teachers were responsible for the coordination and delivery of the intervention activities in four intervention schools. Schools allocated to the comparator arm continued with usual practice.

### Resource Use and Costs

Costs collected focused on the items that were likely to vary between the intervention and control arm. The costs were divided into three categories (**Table 1**) to reflect the process of intervention development, initial "up-front" implementation costs and then any recurring costs associated with delivering the intervention over time.

According to standard practice for economic evaluation, the base case analysis assumed that the intervention was in a "running state" and therefore only costs associated with the ongoing delivery of the intervention were included. To ensure comparability and completeness, the other cost categories (development and up-front initial implementation costs) were reported separately. Sensitivity analysis explored the impact of including the up-front implementation costs, which is detailed later.

First, a public sector perspective was adopted, and all resource use associated with the delivery of the intervention was recorded, using study-specific instruments completed by each CHIRPY DRAGON teacher. This resource use was then multiplied by the relevant unit cost (Yuan currency), obtained from Chinese sources, or valued at market prices, to calculate the total cluster (school)-level cost. These were then averaged across the number of classes and number of children per class ($n$ = 45). Detailed

**TABLE 1** | Cost/Resource use items. Cost-effectiveness of a school-and family-based childhood obesity prevention programme in China: the "CHIRPY DRAGON" cluster-randomised controlled trial, China, 2016–17.

| Category 1: Costs associated with initial development of intervention | Category 2: Initial "up-front" implementation costs | Category 3: On-going delivery/running costs |
|---|---|---|
| • Research staff time for development of the schoolteacher handbook explaining intervention | • Time and travel costs required for the CHIRPY DRAGON teachers to deliver training workshops/sessions | • Labour: CHIRPY DRAGON teachers' time and workshop assistants' time |
| • Hiring of a designer to optimise the presentation of intervention materials (leaflets and illustration media) | • Initial printing of school teacher handbooks | • Intervention materials used during workshops/sessions |
| • Researcher preparation time for CHIRPY DRAGON teacher training | • Time and travel costs related to the set-up meeting to explain about the intervention components to school staff | • Delivery fee for reward boards and loudspeakers |
| • Time and travel costs related to school staff meeting to discuss provision of children's physical activity sessions | • Time and travel costs related to meeting with school catering teams to explain the school lunch improvement objectives | • Office stationery<br>• On-going printing<br>• Incentives: incentive prizes for meeting family healthy behaviour challenges and performance recognition certificates for catering teams<br>• CHIRPY DRAGON teachers' transport<br>• CHIRPY DRAGON teachers' telephone costs |

resource use and unit costs for the public sector perspective are available within the Supplementary File (**Supplementary Table SB**).

Next, a societal perspective was adopted. To capture the cost of school lunch, the catering teams from both the intervention and control schools recorded data on the cost of lunch provision. This data was then adjusted assuming that the cost of provision was passed on fully to the child's family and the 12 months mean estimates used to calculate the incremental cost of lunch within the intervention compared to the control schools. For the intervention schools, family questionnaires also captured data on any family time costs associated with attending the intervention workshops. On average, two family members attended the workshops. To measure the opportunity cost of time, family members were asked what they would have been doing if not attending the workshop. They were asked to select between "work" or "not at work" activities. For the opportunity cost of missed work time, the population average salary was applied [16], and for "non-work" activities the national minimum wage was assumed as a valid cost of leisure time [17].

To aid interpretation, all costs are reported in Chinese Yuan at a 2016–2017 price base and UK Pound/US dollar using Gross Domestic Product Purchasing Power Parities (GDP PPPs) [18].

## Outcomes

All outcomes were collected at the individual level. Assessments were undertaken in each school by independent and trained assessors who were blind to allocation, using standardised procedures and instruments at baseline (start of intervention) and first follow-up (end of intervention) [11]. The primary clinical outcome for effectiveness was the difference in body-mass index (BMI) standard deviation scores (z-scores) between arms at completion of the 12 months intervention. BMI z-scores were calculated using the WHO 2007 Growth Charts [19]. The primary economic outcome measure was quality-adjusted life years (QALYs) and utility-based information to inform the

calculation of QALYs were collected. Utilities were derived from the children using the Chinese version of the CHU-9D (CHU9D-CHN) which was researcher-administered at baseline, and at 12 months [20]. This instrument combines nine dimensions of HRQoL: worried; sad; pain; tired; annoyed; schoolwork/homework; sleep; daily routine; and ability to join in activities [21]. Each dimension comprises five severity levels, resulting in 1,953,125 unique health states associated with the measure. The CHU-9D-CHN instrument has a Chinese tariff set available for estimating utility values, but according to the instrument developers (personal communication), at the time of the study, the Chinese-specific preference weights were still in development and required further validation therefore it was recommended to use the UK tariff set for the primary analysis, and to use the Chinese-tariff set as a sensitivity analysis [22]. Therefore, for the primary analysis, individual responses from the questionnaires were transformed into utility weights derived from a UK general population sample using an algorithm developed by Stevens et al [21]. For the societal perspective, the analysis also included QALY gains/losses falling on adult household members using the validated Chinese version of the EQ-5D-3L instrument [23] and the UK value set. Two family members (main carers) were asked to complete the utility-based questionnaires. The Chinese tariff scores for EQ-5D-3L was also used for comparison in a sensitivity analysis [24]. All QALYs were calculated over the 12 month period, using the standard area under the curve approach [25, 26].

To measure the difference between the intervention and control arm, two mixed linear models were developed, both of which accounted for clustering: a model controlling for baseline outcomes; and a further adjusted model controlling for baseline outcomes and pre-specified covariates including (school-level covariates (i.e., whether the school provides midmorning snack, whether the school has an indoor activity room) and child-level sociodemographic covariates (i.e., age, sex, and mother's education level) and behavioural covariates [daily

average servings of fruit and vegetables, weekly servings of unhealthy snacks and sugar-added drink, objectively measured time in MVPA (minutes/24 h) and objectively measured sedentary time (minutes/24 h)] [11]. The further adjusted model was used in the base case analysis.

## Missing Data

As the public sector perspective only included the costs that fall onto the schools, this resource use was collected at the cluster level (the schools). For all other data, including resource use from the family members and all outcome data, these were collected at the individual level. The reasons for missingness therefore differed between the public and the societal perspective.

For the public sector perspective, there was a very high retention rate and a high level of data completeness (0% missing for the resource use data; less than 4% for the children's outcome data) so there was no need to use multiple imputation methods to account for missing data.

For the societal perspective, although almost 25% of the outcome data for family members were missing, multiple imputation for the primary analysis was not required as all the covariates in the model were fairly complete; and the baseline characteristics of the study participants were well balanced between the two groups [11]. Analyses for both the public sector and the societal perspectives were therefore conducted as intention to treat on randomised participants with available data in STATA version 13.

## Statistical/Economic Evaluation Analyses

Analysis of cost-effectiveness was undertaken according to current best practice methods for conducting economic evaluation alongside cRCT [27]. The cost data was highly skewed therefore a gllamm model was used. The economic analyses took an incremental approach and therefore measured the difference in cost, offset by the difference in outcome measured using BMI (z-score) for the cost-effectiveness analysis, and QALYs for the cost-utility analysis. The results are expressed through the incremental cost-effectiveness ratio (ICER) based on the further adjusted costs and effects. Since a time horizon of 1 year was used, costs and outcomes were not discounted [25].

To account for both the correlation between costs and outcomes (QALYs) and the cluster-design of the trial, the net-benefit regression (NBR) framework was applied [28] to construct the cost-effectiveness acceptability curve (CEAC). The CEAC shows the relationship between the cost-effectiveness (the ICER) and how much society is Willing To Pay (WTP) for a QALY-gain. The NBR puts the analysis of costs and effects into a regression model and computes each child's net benefit (NB) as WTP x $QALYs_i$—$Costs_i$, and then uses the NB as the dependent variable within the model to compute whether the intervention is cost-effective for different levels of WTP for a QALY. For technical details of this approach, please see these additional Refs. [28, 29].

To expand the economic evaluation to a societal perspective and include household spillover effects, all child and family members' QoL data were linked and matched using a published "multiplier" approach [30]. The QALYs were adjusted using the following steps: Step 1: mean incremental QALYs per child calculated (CQ), Step 2: mean incremental QALYs per family member calculated (FQ), Step 3: each child assumed to have two family members in household (n), Step 4: the multiplier for each child was then calculated as:

$$[1 + (nFQ/CQ)]$$

Additionally, an allowance (a figure of around 1.1) was made for the spillovers displaced by the intervention [30]. For further discussion and technical explanation see Al Janabi et al (2016) [30].

The multiplier approach was not applied to the resource use data as it was not possible to link the family-related costs to the individual child. Instead, the costs were simply summed and averaged assuming that each child had at least two family members attend the workshops (as per the intervention protocol). For the societal perspective therefore the base-case ICER was calculated by applying the following formula:

$$\frac{\text{mean incremental public sector costs}}{CQ} * \frac{1.1}{[1 + (nFQ/CQ)]} * \frac{\text{mean incremental societal costs}}{\text{mean incremental public sector costs}}$$

To assess cost-effectiveness, the ICER is compared to a threshold willingness to pay for a unit gain in QALY. As there is no established threshold value for how much society is willing to pay in China, the UK (£20,000) and US (US$50,000) threshold values were used as a reference point [31, 32] as well as the GDP-per capita (US$19,000) thresholds as recommended by WHO [33]. All analyses were conducted in STATA version13.

## Sensitivity Analysis

To assess the robustness of the results to the assumptions made, three sensitivity analyses were undertaken: 1. including initial "up-front" costs associated with implementing the intervention (to test the sensitivity of the results to this increase in costs); 2. using the Chinese tariff value set to estimate QALYs (to test the sensitivity of the results to this change in QALY); and 3. varying the class size by only including consented children (average 41 per class).

An additional sensitivity analysis was applied to the societal perspective that used predictive mean matching to impute the missing data [34]. This was to avoid any loss of efficiency or potential bias of the results with the exclusion of participants with missing data [35, 36].

## RESULTS

## Participants

No schools dropped out of the trial. In total, 1,641 children were recruited and schools randomized to intervention (20 schools, n = 832) and control (20 schools, n = 809) [11]. The baseline characteristics of the study participants were well balanced between the two groups. The mean age of the children was 6.1 years (SD = 0.35) and 54.5% were male. More than a third of parents did not have a university education. Approximately 18% of the children were either overweight (10.8%) or obese

**TABLE 2 |** Costs per child over the 12 months follow-up period from a public sector and societal perspective (Yuan (£/$), 2016/2017 prices). Cost-effectiveness of a school- and family-based childhood obesity prevention programme in China: the "CHIRPY DRAGON" cluster-randomised controlled trial, China, 2016–17.

| Cost item | Mean cost per class Yuan (£/$) | Mean cost per child Yuan (£/$)[a] |
|---|---|---|
| PUBLIC SECTOR PERSPECTIVE (On-going delivery/running costs) | | |
| CHIRPY DRAGON teachers' and workshop assistants' time | 927 Yuan (£183.6/$261) | 20.60 Yuan (£4.08/$5.80) |
| Intervention materials | 95.4 Yuan (£18.9/$26.55) | 2.12 Yuan (£0.42/$0.59) |
| Delivery fee (e.g., for reward boards and loudspeakers) | 1.35 Yuan (£0.27/$0.36) | 0.03 Yuan (£0.006/$0.008) |
| Office stationery | 0.225 Yuan (£0.045/$0.045) | 0.005 Yuan (£0.001/$0.001) |
| Ongoing printing | 272.7 Yuan (£45/$76.5) | 6.06 Yuan (£1.20/$1.70) |
| Incentive prizes | 118.35 Yuan (£23.4/$33.3) | 2.63 Yuan (£0.52/$0.74) |
| CHIRPY DRAGON teachers' transport | 154.35 Yuan (£30.6/$43.2) | 3.43 Yuan (£0.68/$0.96) |
| CHIRPY DRAGON teachers' telephone allowance | 31.95 Yuan (£6.3/$9) | 0.71 Yuan (£0.14/$0.2) |
| Total (Public Sector Perspective) | 1,600.8 Yuan (£317/$449.7) | 35.53 Yuan (£7.04/$10.01) |
| ADDITIONAL COSTS FOR SOCIETAL PERSPECTIVE | | |
| Incremental lunch costs (difference between intervention and control schools) | 5,737 Yuan (£1,136.2/$1,616.4) | 127.5 Yuan (£25.25/$35.92) |
| Parents/main carers' workshop attendance time (two family members) | 16,826.4 Yuan (£3,348/$4,739.4) | 373.92 Yuan (£74.4/$105.32) |
| Total (Societal Perspective) | 24,162.75 Yuan (£4,784/$6,806) | 536.95 Yuan (£106.33/$151.25) |

[a]*Public sector perspective: Data collected at school level (n = 20 intervention schools), there was no missing cost data; assuming average class size of 45.*

(7.1%); comparable to Chinese national data for overweight/obesity in the same age group (20.4%) [37]. There were no differences between the two study groups in completeness of the outcome measures. Overall, BMI z-score was missing for 23 (1.4%) and 60 (3.7%) children at baseline and 12 months follow-up respectively. CHU-9D and EQ-5D-3L utility data were missing for 36 (2.2%) and 54 (3.2%) children and 406 (24.7%) and 415 (25.3%) main carers (parents/grandparents), at baseline and 12 months follow-up, respectively.

## Resource Use and Costs

A detailed breakdown of the resources used for the ongoing delivery of the intervention is available in **Supplementary Tables SB, C**. **Table 2** reports the total mean cost per child and per class over 12 months for both the public sector and societal perspective. Relative to the costs from delivering the intervention, both the development and (in particular) the implementation costs were low (see **Supplementary Tables SD, E** in the supplementary file). The total mean annual cost of lunch per child (based on monthly reporting of lunchtime costs by caterers) was higher in the intervention schools 1,765 Yuan (£349.50/$497.18) compared to the control schools 1,637.5 Yuan (£324.25/$461.26) (see **Supplementary Table SF** within the supplementary file) resulting in an incremental cost of lunch for intervention versus control schools of 127.5 Yuan (£25.25/$35.92) per child over the 12 month trial period (see **Table 2**). A breakdown of the opportunity cost associated with the time spent at the family workshops is presented in the Supplementary File (**Supplementary Table SG**). Overall, 61% of the family members would have otherwise been at paid work. On average, two family members attended the workshops so the total mean opportunity cost of family members' time was 373.92 Yuan (£74.4/$105.32) per child (see **Table 2**).

Overall, the incremental cost associated with the intervention was 35.53 Yuan (£7.04/$10.01) per child and 536.95 Yuan (£106.33/$151.25) per child/family from a public sector and societal perspective, respectively (see **Table 2**, also **Supplementary Table SC** within the supplementary file).

## Outcomes

All outcomes at baseline and 12 months are presented in **Table 3**. QALY and BMI z-score mean differences were 0.004 (0.000–0.007, $p = 0.034$) and −0.13 (−0.26 to 0.00, $p = 0.048$), respectively in the baseline adjusted models, and 0.004 (−0.000 to 0.008, $p = 0.056$) and −0.13 (−0.26 to −0.01, $p = 0.041$) respectively in the further adjusted models. QALY mean difference for parents/grandparents were 0.002 (−0.002 to 0.006, $p = 0.329$) in the baseline adjusted model, and 0.002 (−0.002, 0.007, $p = 0.421$) in the further adjusted model. The QALYs attained were higher for CHU-9D and similar for EQ-5D-3L using the Chinese tariff compared to the UK tariff (Supplementary File, **Supplementary Table SH**). After conducting multiple imputation, the results remain similar to those pre-imputation (Supplementary File, **Supplementary Table SH**).

## Economic Evaluation

From the public sector perspective, the CHIRPY DRAGON intervention cost 272.7 Yuan (£54/US$77) per BMI z-score change and 8,888 Yuan (£1,760/US$2,502) per QALY gained, which is highly cost-effective using all established threshold analyses [31–33] (**Table 4**). Using the UK threshold analysis, the CEAC showed a 96% probability of the intervention being cost effective at a WTP threshold of £20,000 per QALY (**Figure 1**). All three sensitivity analyses did not markedly change these results (**Table 4**).

To account for family member effects, the multiplier for each child was calculated as:

$$[1 + (2(0.002)/0.004)]$$

The base-case ICER was therefore calculated using the following formula:

$$\frac{£7.04}{0.004} * \frac{1.1}{2} * \frac{£106.33}{£7.04}$$

The impact of including the family member QALYs and household costs increased the ICER from 8,888 Yuan (£1,760/US$2,502) to 73,831 Yuan (£14,620/US$20,796) per QALY gained (**Table 4**), but remained cost-effective for all threshold

**TABLE 3** | Outcomes at baseline and 12 months. Cost-effectiveness of a school-and family-based childhood obesity prevention programme in China: the "CHIRPY DRAGON" cluster-randomised controlled trial, China, 2016–17.

| Outcomes | Control group | | Intervention group | | Adjusted mean (95% CI) | | | |
|---|---|---|---|---|---|---|---|---|
| | N | Mean (SD) | N | Mean (SD) | Difference[a] (Intervention vs control) | p-value | Difference[b] (Intervention vs control) | p-value |
| Baseline | | | | | | | | |
| BMI z-score | 796 | −0.13 (1.30) | 822 | −0.13 (1.30) | | | | |
| CHU-9D utility | 793 | 0.936 (0.069) | 812 | 0.938 (0.068) | | | | |
| EQ-5D-3L utility | 596 | 0.961 (0.085) | 639 | 0.962 (0.081) | | | | |
| At 12 months | | | | | | | | |
| BMI z-score | 777 | −0.23 (1.34) | 804 | −0.35 (1.22) | −0.13 (−0.26 to 0.00) | 0.048 | −0.13 (−0.26 to -0.01) | 0.041 |
| CHU-9D (QALYs for children) | 781 | 0.932 (0.067) | 806 | 0.937 (0.059) | 0.004 (0.000–0.007) | 0.034 | 0.004 (−0.000–0.008) | 0.056 |
| EQ-5D-3L (QALYs for family members) | 584 | 0.965 (0.061) | 642 | 0.966 (0.066) | 0.002 (−0.002–0.006) | 0.329 | 0.002 (−0.002–0.007) | 0.421 |

*Notes: BMI, body mass index; CHU-9D, Child Health Utility 9D; CI, Confidence Interval; EQ-5D-3L, Euro-QoL instrument; QALYs, Quality-Adjusted Life Years; SD, Standard Deviation.*
*[a]=baseline adjusted model: adjusted for school clustering and baseline outcome.*
*[b]=further adjusted model: adjusted for baseline outcome, prespecified school-level (i.e., whether the school provides midmorning snack, whether the school has an indoor activity room) and child-level sociodemographic (i.e., age, sex, and mother education level) and behavioural [daily average servings of fruit and vegetables, weekly servings of unhealthy snacks and sugar-added drink, objectively measured time in MVPA (minutes/24 h) and objectively measured sedentary time (minutes/24 h)] covariates.*
*Base case analysis used the further adjusted model.*

**TABLE 4** | Base case and sensitivity analysis results for both public and societal perspective. Cost-effectiveness of a school-and family-based childhood obesity prevention programme in China: the "CHIRPY DRAGON" cluster-randomised controlled trial, China, 2016–17.

**PUBLIC SECTOR PERSPECTIVE**

| Analysis | ICER (cost per additional QALY) |
|---|---|
| Primary (base case) analysis | 8,888 Yuan (£1,760/US$2,502) |
| Sensitivity analysis 1: implementation costs included | 9,760 Yuan (£1,922/$US2,732) |
| Sensitivity analysis 2: using Chinese value set | 5,923 Yuan (£1,173/US$1,668) |
| Sensitivity analysis 3: class size *n* = 41 | 9,756 Yuan (£1,932/US$2,742) |

**SOCIETAL PERSPECTIVE**

| Analysis | ICER (cost per additional QALY) |
|---|---|
| Primary (base case) analysis | 73,831 Yuan (£14,620/US$20,796) |
| Sensitivity analysis 1: implementation costs included | 74,280 Yuan (£14,709/US$20,923) |
| Sensitivity analysis 2: using Chinese value set | 73,346 Yuan (£14,524/US$20,660) |
| Sensitivity analysis 3: class size *n* = 41 | 81,037 Yuan (£16,047/US$22,823) |
| Sensitivity analysis 4: using predictive mean matching to impute missing data | 84,380 Yuan (£16,709/US$23,767) |

analyses, apart from when 1xGDP per capita threshold (US$19,000) was applied [31–33]. The ICER did not change substantially in all four sensitivity analyses (maximum of 84,380 Yuan (£16,709/US$23,767) per QALY when predictive mean matching multiple imputation was applied) (**Table 4**).

## DISCUSSION

This is the first study to present a robust economic evaluation of a multi-component school-based childhood obesity prevention programme in a Chinese setting. It has shown that from a public sector perspective and a societal perspective this intervention is cost-effective and that this result was robust to all sensitivity analyses.

### Strengths and Limitations of This Study

A particular strength of the study was the large sample size (1,641 children), standardised data collection procedures, and the low level of attrition throughout the follow up period. For the analysis at the public sector perspective, there was a low level of missing data. This study reported the ICER from two alternative perspectives and included both clinical and economic outcomes. This enabled comparison with other studies. Furthermore, this study is one of the very few economic evaluations of obesity prevention studies worldwide and the first in China, which measured QALYs in children as young as 6 years and included family member effects. It used both the UK and Chinese tariffs for deriving the QALYs. Moreover, this is the first economic evaluation study worldwide to consider health spillover effects generated from a behavioural obesity intervention using a multiplier approach. The intuition behind the multiplier approach is that there is a bigger health dividend for the population than is represented just by children's QALYs and therefore this wider health dividend should be captured within an economic evaluation [30].

The study also had some limitations. One potential limitation relates to the way HRQoL information was collected from the children. There may have been an

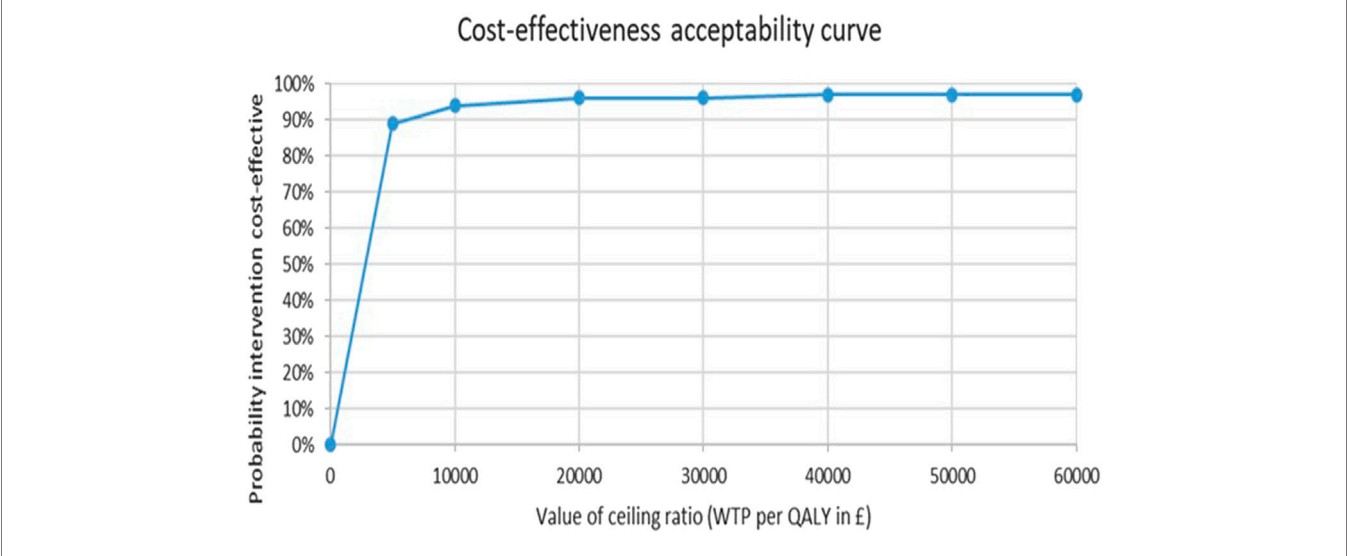

**FIGURE 1 |** Cost-Effectiveness Acceptability Curve. Cost-effectiveness of a school-and family-based childhood obesity prevention programme in China: the "CHIRPY DRAGON" cluster-randomised controlled trial, China, 2016–17.

influence on how children completed the questionnaire as items and possible responses within the CHU-9D were read to the children. This could have led to responder-bias [38]. However, given the young age of the participants, this collection strategy was chosen to optimise data quality and completion. Furthermore, interviewers were blind to allocation, minimising any differential bias. A further limitation was that the response rate was low from family members. This may result in a lack of power to detect significant effects in family members. Both children's and family members' incremental QALYs were estimated separately before aggregating the mean estimates. In future studies, where the number of children and household members are more similar, we would recommend using a dyadic approach. The advantage of dyadic analysis, compared to the multiplier approach for including health spillovers is that it enables a probabilistic sensitivity analysis to be conducted to explore uncertainty. The analysis was also limited to the time horizon of the intervention and therefore the sustainability of effect remains unknown. Finally, as this is a behavioural intervention, therefore highly dependent on cultural, infrastructural and other system-related aspects, the generalisability of results to other contexts, particularly to other country settings, needs to be explored further [39].

## Comparison With Other Studies

The most recent systematic review of economic evaluations [1] of obesity prevention interventions, which was limited to children and adolescents, found it challenging to synthesise the studies due to the heterogeneity of outcome measures used and the lack of an acceptable WTP threshold for a "weight" outcome. However, it suggested that all school-based obesity interventions appear cost-effective, and this study adds to the evidence base as outcomes

have been measured using QALYs, a consistent outcome measure which aids comparison.

The cost-effectiveness analysis within this study led to an ICER from a public sector perspective of 272.7 Yuan per BMI z-score change. This was lower than two previous trial-based intervention studies which used BMI z-score as their measure of effectiveness: one Chinese study, targeting dietary habits and physical activity in children 6–13 years, (885 Yuan per BMI z-score change) [7]; the other Australian, targeting physical activity in adolescents 13–16 years, (1,537 Yuan per 10% reduction in BMI z-score) [40]. Neither of these studies included indirect costs. Contextual factors including differences in the stage of the childhood obesity epidemic and cultural factors, as well as intervention differences (e.g., target, components and how these were delivered), may contribute to differences in findings. It has previously been determined that obesity prevention interventions are more effective when delivered by dedicated staff rather than classroom teachers [11] and the CHIRPY DRAGON staff employed in this trial were well accepted by schools and their costs were included. The use of dedicated delivery staff helped to maximise the consistency and quality of implementation as school teachers are often overloaded and struggle to find capacity for delivery.

## Conclusion

The results from this study demonstrate the cost-effectiveness of the CHIRPY DRAGON intervention from both a public sector and a societal perspective and will inform obesity prevention policy in China as well as in other country settings, as well as highlighting future research needs. Long term monitoring needs to be put in place to assess sustainability of effect so that a greater understanding of long-term cost-effectiveness can be obtained.

# ETHICS STATEMENT

The studies involving human participants were reviewed and approved by the Life and Health Sciences Ethical Review Committee at the University of Birmingham (2nd March 2015) and the Ethical Committee of Guangzhou Centre for disease Control and Prevention (1st December 2014). Written informed consent to participate in this study was provided by the participants' legal guardian/next of kin.

# AUTHOR CONTRIBUTIONS

MZ: economic analysis, methodology, writing the original draft, reviewing and editing. PA: conceptualization, investigation, methodology, supervision, reviewing and editing. BL: conceptualization, funding acquisition, investigation, methodology, project administration, supervision, reviewing and editing. MP: investigation, methodology, resources, reviewing and editing. WL: investigation, project administration, resources, reviewing and editing. KH and JM: methodology, resources, reviewing and editing. RL and WL: investigation, reviewing and editing. KC: conceptualization, investigation, methodology, reviewing and editing. EF: methodology, investigation, supervision, resources, reviewing and editing.

# FUNDING

This study was funded through a charitable donation from Zhejiang Yong Ning Pharmaceutical Ltd. Co. (trial registration number: ISRCTN11867516) from 2014 to 2018 to the University of Birmingham. The funder played no role in study design, implementation, analysis and reporting. The views and opinion expressed therein are those of the authors and do not necessarily reflect those of the funders. The analyses were further supported by a University of Birmingham College of Medical and Dental Sciences PhD studentship.

# CONFLICT OF INTEREST

PA, EF, MP, KC, KH, and JM report grants from the NIHR on unrelated research. PA, EJ, MP, KH, and KC report grants from the NIHR related to research on childhood obesity prevention. EF and PA were trustees of the Association for the Study of Obesity (PA 2014–2017; EF 2017–2019).

The remaining authors declare that the research was conducted in the absence of any commercial or financial relationships that could be construed as a potential conflict of interest.

# ACKNOWLEDGMENTS

We would like to thank the schools, parents and children who participated in the study and for the Chinese Local Authorities for their support. We would also like to thank Hareth Al-Janabi, and Raymond Oppong at the University of Birmingham for their guidance with the economic evaluation methods and analysis.

# SUPPLEMENTARY MATERIAL

The Supplementary Material for this article can be found online at: https://www.ssph-journal.org/articles/10.3389/ijph.2021.1604025/full#supplementary-material

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
