## [Reviewer comments · International Journal of Public Health]

Peer Review Report

Review Report on Cost-effectiveness of a school-and family-based childhood obesity prevention programme in China: the 'CHIRPY DRAGON' cluster-randomised controlled trial

Original Article, Int J Public Health

Reviewer: Reviewer 1

Submitted on: 01 Jun 2021

Article DOI: 10.3389/ijph.2021.1604025

EVALUATION

Q 1 Please provide your detailed review report to the authors. The editors prefer to receive your review structured in major and minor comments. Please consider in your review the methods (statistical methods valid and correctly applied (e.g. sample size, choice of test), is the study replicable based on the method description?), results, data interpretation and references. If there are any objective errors, or if the conclusions are not supported, you should detail your concerns.

This is a well written paper on an important subject.

My major comment is that the economic evaluation of Chirpy Dragon has already been reported in Li et al, Plos Medicine 2019. Similarly Figure 1 has already been published, so the content is not new. The extra contribution of the current manuscript above the PLOS Medicine paper is the detailed costing which is very hard to follow.

Major comments

Please provide a CHEERs checklist.

Please provide intervention costs in the body of the manuscript - these could be added to Table 1. It would be helpful to see the mean cost per participant .

Please justify why intervention development is included in the

Why was the Chinese valuation of the CHU9D not used in the primary analysis?

Sensitivity analyses are not presented transparently, but are referred to. Please provide in supplementary results a table with sensitivity analyses investigated and impact on ICERs

Minor comments

In Results line 176 states incremental cost associated with the intervention is 35.53 Yuan - please reference the table where the reader can see this amount.

Similarly line 189 - ICERs are cited - please refer to the table where these results can be found

Lines 213-215 describes an approach to include spillover effects - this should be in the methods

Q 2 Please summarize the main findings of the study.

A school delivered obesity prevention programme in China was found to be cost-effective

Q 3 Please highlight the limitations and strengths.

Strengths - taking both a public sector and societal perspective; carrying out sensitivity analysis; inclusion of spillover effects.

Limitations- presentation of costs only in the supplementary appendix; difficulty in following where costs are presented

PLEASE COMMENT

Q 4 Is the title appropriate, concise, attractive?

Yes

Q 5 Are the keywords appropriate?

yes

Q 6 Is the English language of sufficient quality?

Yes

Q 7 Is the quality of the figures and tables satisfactory?

No.

Q 8 Does the reference list cover the relevant literature adequately and in an unbiased manner?)

yes

QUALITY ASSESSMENT

Q 9 Originality

Q 10 Rigor

Q 11 Significance to the field

Q 12 Interest to a general audience

Q 13 Quality of the writing

Q 14 Overall scientific quality of the study

REVISION LEVEL

Q 15 Please take a decision based on your comments:

Major revisions.